# Optimization of Oxford Nanopore Technology Sequencing Workflow for Detection of Amplicons in Real Time Using ONT-DART Tool

**DOI:** 10.3390/genes13101785

**Published:** 2022-10-03

**Authors:** Robert Player, Kathleen Verratti, Andrea Staab, Ellen Forsyth, Amanda Ernlund, Mihir S. Joshi, Rebecca Dunning, David Rozak, Sarah Grady, Bruce Goodwin, Shanmuga Sozhamannan

**Affiliations:** 1Applied Physics Laboratory, The Johns Hopkins University, Laurel, MD 20723, USA; 2Datirium, LLC, Cincinnati, OH 45226, USA; 3Naval Surface Warfare Center, Dahlgren, VA 22448, USA; 4US Army Medical Research Institute of Infectious Diseases (USAMRIID), Fort Detrick, MD 21702, USA; 5Joint Program Executive Office for Chemical, Biological, Radiological and Nuclear Defense (JPEO-CBRND), JPL-Enabling Biotechnologies, Frederick, MD 21702, USA; 6Logistics Management Institute, Tysons, VA 22102, USA

**Keywords:** biodefense, biodetection, biosurveillance, biothreat agents, oxford nanopore sequencing, real-time Sequencing, sequencing library preparation

## Abstract

An optimized, well-tested and validated targeted genomic sequencing-based high-throughput assay is currently not available ready for routine biodefense and biosurveillance applications. Earlier, we addressed this gap by developing and establishing baseline comparisons of a multiplex end-point Polymerase Chain Reaction (PCR) assay followed by Oxford Nanopore Technology (ONT) based amplicon sequencing to real time PCR and customized data processing. Here, we expand upon this effort by identifying the optimal ONT library preparation method for integration into a novel software platform ONT-DART (ONT-Detection of Amplicons in Real-Time). ONT-DART is a dockerized, real-time, amplicon-sequence analysis workflow that is used to reproducibly process and filter read data to support actionable amplicon detection calls based on alignment metrics, within sample statistics, and no-template control data. This analysis pipeline was used to compare four ONT library preparation protocols using R9 and Flongle (FL) flow cells. The two 4-Primer methods tested required the shortest preparation times (5.5 and 6.5 h) for 48 libraries but provided lower fidelity data. The Native Barcoding and Ligation methods required longer preparation times of 8 and 12 h, respectively, and resulted in higher overall data quality. On average, data derived from R9 flow cells produced true positive calls for target organisms more than twice as fast as the lower throughput FL flow cells. These results suggest that utilizing the R9 flowcell with an ONT Native Barcoding amplicon library method in combination with ONT-DART platform analytics provides the best sequencing-based alternative to current PCR-based biodetection methods.

## 1. Introduction

Since its first inception more than two decades ago, next generation sequencing (NGS) technologies have led to advancements in multiple fields including environmental surveillance and variant detection, as highlighted during the response to the SARS-CoV-2 pandemic [1,2]. However, while sequencing-based approaches offer many advantages over traditional molecular approaches such as PCR, its widespread use and acceptance in clinical diagnostics lags. This is likely due to a number of factors including cost and the complexity of generating, analyzing, and adjudicating sequencing results in a timely manner to make actionable calls either in clinical (treatment decision) or biosurveillance/biodefense (risk mitigation) scenarios. There is also a bottleneck in adopting sequence-based biothreat detection in far-forward or resource limited settings.

One approach that has been explored is the use of amplicon sequencing (AmpSeq), as this strategy includes an initial PCR step, which is both well-accepted by the field and has historical data to assess feasibility and performance. This approach is also less complicated from an analytics point of view than a metagenome sequencing approach (MetaSeq), wherein everything in a sample is sequenced. However, in the MetaSeq approach, lack or loss of strain-level specificity makes adjudication of an actionable call significantly more challenging. The advantages of AmpSeq over real time PCR assays includes both higher throughput of samples and the higher target multiplexing capability per sample. The sequencing approach has been accelerated by Nanopore-type technologies that require minimal sequencing infrastructure and the ability to sequence anything, anywhere, and anytime by virtually anyone. Even with these clear advantages, to date there have been only a handful of studies investigating the feasibility of AmpSeq or MetaSeq approaches in biodefense applications [3,4,5,6,7,8,9].

In one of these recent studies [10], a multiplexed ONT AmpSeq protocol using a ligation-based library preparation was found to be a viable alternative to existing singleplex or limited multiplex (up to 14 targets) PCR assays for detecting biothreat agents. While promising, this work also suggested that in order to reliably detect the target amplicon regions of the genome, an initial PCR amplification step was necessary prior to sequencing, even when high levels of the target genetic material was present in the sample. Additionally, the protocol required nearly 14 hours, limiting its long-term utility in resource-limited settings [10].

To directly address these pitfalls, in the current study we compared four different library preparation protocols to identify which produces the best balance of high-quality sequence data with the shortest time from sample-to-answer. The protocols evaluated included: ligation-based (LIGTN), native barcoding-based (NATBC), and two 4-primer options (4-primer ONT-4PONT and 4-primer standard-4PSTD). In order to compare the data from these sequencing protocols appropriately and reproducibly, an alignment pipeline was implemented and ultimately built upon to produce a novel software application called ONT-DART (Oxford Nanopore Technology- based Detection of Amplicons in Real Time). This application has a graphical user interface, has been optimized to run on a portable sequencing device (the MK1C), and contains multiple steps aimed at reducing false negative and positive detection events. Here, we present a comparative analysis of four sequencing library preparation methods and discuss how ONT-DART is used to process real-time sequence data and make actionable calls.

## 2. Materials and Methods

### 2.1. Sample Preparation and PCR Amplification

In most cases, bacteria were obtained from the USAMRIID’s Biodefense Reference Material Repository. DNA was extracted from each bacterial strain using the Qiagen EZ1 DNA Tissue Kit (Qiagen, MD, USA; Catalog#: 953034). Each strain was cultured on solid media prior to extraction. A portion of the cultured material was suspended in G2 lysis buffer and loaded onto the Qiagen EZ1 instrument. For BAP708 specifically, 10 µL of lysozyme was added to the lysis suspension, and the suspension was incubated for 30 min at 37 °C, shaking at 1000 rpm prior to loading onto the instrument. The extraction was performed on the instrument using the Qiagen EZ1 DNA Bacteria Card (Qiagen, Gaithersburg, MD, USA). The DNA extracts were quantified using a Qubit v3 instrument with the dsDNA HS Assay Kit (Thermo Fisher Scientific, St. Louis, MO, USA; Catalog #: Q32854). The concentrations of the extracted DNA samples were adjusted with nuclease-free water (Ambion, Austin, TX, USA; Catalog#: AM9937) to yield 100 µL of DNA at a concentration of 20 ng/µL.

A total of 10 ng of gDNA resulting from the above-described extraction process was added to a 23-plex PCR cocktail and amplified to generate the amplicons for the ligation and native-barcoding sequencing libraries. Details of the amplification process are described in Player et al. [10] (see Appendix A for amplicon details).

### 2.2. Library Preparation Method 1: 4PSTD

A modified version of the 4P protocol (Four-primer PCR SQK-PBK004, version: FFP_9038_v108_revQ_14Aug2019) was carried out as described by the manufacturer (Oxford Nanopore Technologies, New York, NY, USA), but with the omission of the AMPure XP bead clean-up steps to reduce sample processing time. Briefly, an up-front amplification (using a 23-plex primer cocktail) and barcoding step was performed in 50 µL reaction volumes containing 50 nM Custom Forward and Reverse Primers, 1.5 µL of 10 µM ONT Barcoded Primers, and 25 µL 2× LongAmp Hot Start Taq Master Mix. Template material of 40 ng was also added to the Master Mix to bring the total to 50 µL. Cycling parameters were: 50 °C for 2 min, 95 °C for 20 sec, followed by 45 cycles of 95 °C for 3 s, 60 °C for 30 s, then holding indefinitely at 4 °C. Equal volumes of barcoded samples were pooled together and brought to 100 fmol in 10 µL water according to the ONT protocol (input DNA/RNA QC Version: IDI_S1006_v1_revB_18Apr2016). This was incubated at room temperature for 5 min with 1 µL of RAP enzyme from the SQK-PBK004 sequencing kit. The remainder of the protocol was followed according to the manufacturer’s instructions.

### 2.3. Library Preparation Method 2: 4PONT–4-Primer Modified

A modified version of the 4P protocol was carried out as described by the manufacturer using a different set of custom cycling conditions that are used in the current PCR field applications. All reactions contained the same components as described for the 4PSTD protocol, but cycling parameters were modified to: 94°C for 1 min, followed by 45 cycles of 94 °C for 30 sec, 60 °C for 30 s, 64 °C for 50 s, then holding indefinitely at 4 °C.

### 2.4. Library Method 3: NATBC–Native Barcoding

All steps were followed according to the Native barcoding amplicons ONT protocol (EXP-NBD104, EXP-NBD114, and SQK-LSK109, version: NBA_9093_v109_revH_12Nov2019). Briefly, amplicons were end-prepped with the Ultra II End-prep kit, incubated at 20 °C for 5 min and 65 °C for 5 min, and cleaned using a 1:1 ratio of AMPure XP beads. Barcodes were ligated using the Blunt/TA Ligase and incubated for 10 min at room temperature followed by an additional 1:1 ratio AMPure XP (Beckman Coulter, Indianapolis, USA) Bead clean-up. Equal volumes of barcoded samples were pooled and Adapter Mix II (Oxford Nanopore Technologies, New York, NY, USA) sequences were added via Quick T4 DNA ligase (New England Biolabs, Ipswich, MA, USA) and incubated at room temperature for an additional 10 min. A final AMPure XP bead clean-up was performed prior to loading the completed libraries on to flow cells.

### 2.5. Library Method 4: LIGTN-Ligation

All steps were followed according to the PCR barcoding (96) Oxford Nanopore Technologies, New York, NY, USA) genomic DNA ONT protocol (SQK-LSK109, version: PBGE96_9068_v109_revG_23May2018). Briefly, PCR barcodes were attached to amplicons using LongAmp Taq 2x master mix (New England Biolabs, Ipswich, MA, USA) with the following cycling conditions: 95 °C for 3 min, followed by 15 cycles of 95 °C for 15 s, 62 °C for 15 s, and 65 °C for 30 s, and a final 2 min incubation at 65 °C totaling approximately 2 h including set up and PCR amplification. Equal volumes of barcoded samples were pooled, end-prepped with the NEBNext FFPE DNA Repair kit (New England Biolabs, Ipswich, MA, USA) and Ultra II End-prep kit, (New England Biolabs, Ipswich, MA, USA) and cleaned with AMPure XP beads totaling approximately 4 h. Sequencing adaptors were ligated and the library underwent a final AMPure XP bead clean-up totaling 1 h. All steps combined, the ligation library method took approximately 7 h to complete.

### 2.6. ONT Sequencing

Finished libraries were quality checked using the Agilent 2200 Tape Station System (Agilent Technologies, Santa Clara, CA, USA) and software version a.02.02 and the Qubit 4 Fluorimeter (Thermo Fisher Scientific, St. Louis, MO, USA). Sample libraries were then sequenced using the Oxford Nanopore GridION, MinION, or MK1C instruments (Oxford Nanopore Technologies, New York, NY, USA) with a data collection period of 24 h. The manufacturer’s guidelines were followed when loading libraries onto both the FL and R9 flow cells. A total of 30 µL was loaded onto the FL flow cell, which included 15 µL of sequencing buffer, 10 µL of loading beads, and 5 µL of prepared DNA library. A total of 75 µL was loaded onto the R9 flow cell, which included 37.5 µL of sequencing buffer, 25.5 µL of loading beads, and 12 µL of prepared DNA library.

### 2.7. Post-Sequence Processing and Analysis

Raw ONT signal data were base called using “fast basecalling” mode in MinKNOW (Release 21.02.1), de-multiplexed, and trimmed using default parameters within the guppy workflow (guppy_basecaller Version 2.3.5 + 53a111f and guppy_barcoder Version 2.3.5 + 53a111f). The following describes the primary steps of the backend processing for the ONT-DART analysis platform. Intermediate files were then used to generate publication-specific results. The “fastq_pass” directory output from MinKNOW was used as input into the ONT-DART pipeline. Each FASTQ (multiple per barcode) is then aligned to an amplicon reference database using BLASTN (v2.6.0+) [11] and filtered to include only those having an identity ≥90% and ≥90% alignment length to the reference (parameters “blastn -num_threads 1-db “$REF” -query <(sed-n ‘1~4s/^@/>/p;2~4p’ “$1” 2 >/dev/null) -outfmt 6”, the sed command formats the FASTQ read input into FASTA format required for blastn queries, and $REF is the variable containing the file path to the amplicon reference blast indexed database). If more than one alignment was reported for an individual sequence, only the highest bitscore alignment was retained for counting. Outside of ONT-DART, sequences were also binned and then analyzed by output time using the custom BASH script nanotimeparse (https://github.com/raplayer/nanotimeparse.git, accessed on 1 September 2021) that utilizes GNU Parallel and GNU core utilities [12]. Amplicon count data figures were generated using ggplot2 in the R Project for Statistical Computing software [13,14].

For real time analyses using the ONT-DART platform there are four required inputs: (i) the full filepath to the “fastq_pass” folder output from MinKNOW during a sequencing run, (ii) the three barcodes used for NTCs, (iii) total number of threads to utilize during processing, and (iv) the number of seconds between each analysis interval. At each analysis interval, only newly generated FASTQ files from all barcodes of the sequencing run are symlinked and processed at that time. The counts from previous intervals are aggregated with the new data, so these reads will not need to be reprocessed at each interval saving significant processing time. If the analysis interval is longer than the time it takes to process new reads, the remainder of the interval time (interval time minus processing time) will elapse before running the next analysis. Otherwise, the next analysis will begin immediately. The ONT-DART software has a single primary heatmap visualization of aligned read count data as its output that has a toggle for displaying whether the organism is detected or only a subset of associated amplicons. Additionally, there are three tables below the heatmap that include a list of detected organisms among the sequenced samples, specific aligned read count information per amplicon, and a summary of aligned reads among NTC samples. For detailed output descriptions and examples, download, and installation instructions please visit our public github repository.

## 3. Results and Discussion

### 3.1. Description of Amplification Strategies

Four amplification/sequencing library preparation protocols were compared for their ability to generate high quality sequencing data: the 4-primer approach as described by the ONT protocol (4PONT), the same 4-primer approach using alternative PCR conditions (4PSTD), the ligation-based approach used in previous work (LIGTN), and the native barcoding approach (NATBC). A list of critical steps and approximate “hands on” preparation times for each protocol are shown in Table 1 and are detailed in Materials and Methods. Generalized workflows are shown in Figure 1.

In the Ligation and Native Barcoding experiments, 10 ng of genomic material was added per reaction to a multiplexed PCR cocktail. In the 4-primer experiments, 40 ng of genomic material was added per reaction. The PCR primer cocktail included the 14 assays described in Player et al. [10], with an additional 9 assays. The list of 12 organisms from which genomic materials were derived along with their associated target assays from the 23-plex are shown in Table 2.

### 3.2. Comparison of Library Preparation Strategies

A total of 384 samples were processed and analyzed on 32 flow cells in order to evaluate these protocols. The test matrix included biological triplicates of each of the 12 organisms prepared using each of the 4 library preparation techniques. A single R9 or FL flow cell contained 3 pooled organisms prepared using the same library preparation protocol. Triplicate no-template controls (NTC) processed through the same library preparation method were included in each flow cell, bringing the total barcoded samples per flow cell to 12.

Flow cell and library preparation comparisons were made using 6 metrics: (i) total reads having a PHRED quality score ≥ 7 (fastq_pass), (ii) aligned read counts, (iii) number of organisms successfully detected, (iv) the difference between the minimum true positive (TP) read count and maximum false positive (FP) read count for each assay after 24 h of sequencing and data collection, (v) proportion of aligned reads identified as FP after 24 h of sequencing and data collection, and (vi) the number of TPs and FPs at each 5 min interval over the first hour of sequencing and data collection. Metrics (i–iv) provide a general summary of sequencing fidelity and “actionable” endpoint calls, while metric (v) helps determine an appropriate proportional read count threshold for calling a positive in an unknown sample. Metric (vi) aims to determine whether TP and FP rates in data generated over short time intervals remain consistent, in the expectation of achieving a high confidence answer minutes after loading a lower-cost, lower-throughput FL flow cell.

Read counts for each sample produced from each library method run for 24 h on the two flow cell types are displayed in Figure 2 (all read count and alignment and detection data is presented in Appendix A). Samples are separated into two groups; samples with spiked organism gDNA (group 0) and NTCs (group 1). Figure 2A shows that the median total reads generated in both 4-primer methods (4PONT and 4PSTD) from NTC samples were greater than that from spiked samples, suggesting the barcoded PCR primers did not amplify efficiently. The barcoded libraries were pooled together in equal volumes and DNA concentrations were not normalized or adjusted prior to sequencing which also could have contributed to the low read counts. However, this trend is reversed following alignment and filtering of reads to the amplicon reference database (Figure 2B). The LIGTN and NATBC samples showed greater median total reads for spiked samples than NTC samples both before and after alignment. This is especially apparent in the NATBC method, and is further resolved in Figure 2C, which shows the aligned read counts as a proportion of the total.

A comparison of the number of successfully detected spiked organisms per library preparation and flow cell is shown in Figure 3. Here, detection is defined as having at least one filtered and aligned read to each of the reference amplicons associated with the spiked genetic material for the sample. This analysis suggests that the NATBC libraries run on the R9 flow cell have the highest frequency of detection.

In order to determine how individual assays from the 23-plex cocktail performed following each library preparation strategy, the difference between the minimum TP and maximum FP aligned filtered read count per amplicon was calculated (Figure 4). Here, a larger value indicates a greater spread between these two values and lower probability of returning a FP detection. Again, samples prepared using the NATBC method and run on the R9 flow cell performed best overall. The other methods have values closer to zero, meaning the probability of these methods producing a FP call is higher than samples prepared with NATBC and run on an R9 flow cell.

The proportion of aligned reads identified as FP after 24 h of data collection for each library preparation-flow cell combination are shown in Figure 5. The two 4-Primer methods return lower FP rates than the LIGTN or NATBC methods; however, this advantage is negated by the higher number of false negatives (Appendix A). These false negatives could be the result of the significantly higher barcode oligo concentration at the amplification step relative to the gene-specific primers, effectively diminishing target binding. These concentrations were derived directly from the ONT 4P protocol and could be leveraged to optimize performance. Although there are 6 samples showing a FP rate of approximately 0.75 for the NATBC method on R9 flow cell (Figure 5A), the remaining 30 samples have <0.02 FP aligned read proportion (Figure 5B). This type of analysis allows for the potential assignment of a percent read count threshold to eliminate FP calls for any given library preparation-flow cell combination.

When analyzing reads produced during the first hour of sequencing using either flow cell, the NATBC method provided successful TP detections in the shortest time for the most organisms, followed by the LIGTN method (Table 3). The threshold for a positive call was defined here as >9 reads aligning to the associated amplicon(s) in the reference database. By this measure, neither 4-Primer method returned a positive result within the first hour of sequencing for any tested organism. While both the NATBC and LIGTN libraries returned positive results for some tested organisms after approximately 5 min of sequencing time on the R9 flow cell, the same calls required slightly more time when using the FL, and detection from the NATBC samples were returned approximately 15 min prior to the LIGTN samples. Figure 6 shows the first hour of data for FRAN239. Here, it is clear that the NATBC method yields the greatest spread between all TP amplicons and any other spurious detections from unexpected amplicons (FPs), while other methods either have overlapping TP and FP counts (LIGTN) or only one of the expected amplicons being detected (4PONT and 4PSTD). These plots for all other tested organisms may be found in Appendix A.

### 3.3. Analysis of NTCs

Three NTCs were included per flow cell for all experiments described above. The mean, median, and standard deviation of the percent aligned reads among NTC samples, as well as the associated percent FP aligned reads among test samples per flow cell was calculated and are shown in Table 4. The two 4-primer methods have a 0.00% mean and median percent aligned reads among NTCs for both FL and R9. However, the non-zero mean percent FP aligned reads among a majority of test samples prepared by these methods indicates that there is either contamination or off-target priming occurring that cannot be accounted for in these internal NTC data. The LIGTN and NATBC methods have non-zero mean percent aligned reads for both NTC samples and FP aligned reads of test samples. This implies that there is the potential for suppression of TP calls when using our method, as well as similar analysis tools like DETEQT [4].

### 3.4. ONT-DART Analysis Pipeline

With the data produced in the experiments described above, a reproducible analysis pipeline was required in order to appropriately compare the library preparation methods. This initial pipeline was a basic wrapper around the gold standard alignment algorithm BLASTN, and the output from this is what is presented in the tables and figures above. In order to leverage the real-time nature of the ONT platform, it was then necessary to further modify and build upon this initial wrapper. Ultimately, this resulted in the real-time enabled analysis software called ONT-DART (Appendix A), which also applies various thresholds for the determination of a positive detection of each amplicon in a sample, and subsequently each organism that is represented by one or more of these target amplicons (Table 2). Thresholds are applied in three steps from the most granular aspect of a sequencing run to least; per read, per sample, and finally, per flow cell. Within this pipeline, each read output from a flow cell was base called, binned per barcode, and then aligned to a set of amplicon reference sequences. Each barcode could have reads aligning to multiple amplicon reference sequences, and each flow cell could have up to twelve barcoded samples, three of which (for our application) must be NTCs.

Using this novel analytical pipeline, the per-read alignment threshold was applied first. Each base called read having an average PHRED quality score ≥ 7 (these are reads output to the “fastq_pass” directory of the MinKNOW output, which is then used as an input to ONT-DART) was aligned to a set of amplicon references using blastn. If there were multiple alignment hits output for a read, only the best alignment by identity was kept for further filtering. Reads having their best alignment identity greater than or equal to 90% and having an alignment length of at least 90% of the length of the reference amplicon sequence were counted toward the total aligned read count for that amplicon.

After the first thresholding step was completed, each barcoded sample in a pooled library would have a read count per amplicon reference sequence in the blastn database index. The sum of these read counts would provide the total aligned read count of the sample. To apply the per sample thresholding, an amplicon reference was only considered positive if its read count is at least 2.0% of the total aligned read count of the sample. This percent was based on two observations: (i) a median false positive rate of 1.50% and 0.06% for the Native Barcoding library preparation method for the FL and R9 (Table 4), respectively, and (ii) the extremely unlikely scenario of having 50 or more amplicon targets truly present in a single sample. This second observation is for targets in a sample, and does not limit the number of PCR assays that may be in the PCR multiplex cocktail, as this cocktail limit is theoretically much greater.

The last applied threshold takes into consideration NTC aligned read counts per flow cell. Each pooled library containing 12 barcoded samples run on a flow cell included three barcoded NTCs. The samples contain only water or buffer and are run through the same library preparation method as the test samples. NTC thresholds for each amplicon reference were unique to each flow cell, and were equivalent to the mean NTC aligned read count for that amplicon plus three times the standard deviation of this metric. This was derived from the concept underlying the DETEQT pipeline previously described, which utilizes multiple negative controls in a similar manner [4].

## 4. Conclusions

In this study, the NATBC protocol on an R9 flow cell coupled with the ONT-DART platform for data processing, filtering, and analysis provided the best sequencing-based alternative to current PCR-based biodetection methods. In this study, two 4-Primer based library preparation strategies were found to require the least amount of time for completion but produced low fidelity data that was not sufficient for target identification using either ONT flow cell type. The best performing library preparation-flow cell combination strategy was found to be the NATBC (Native Barcoding) method on the R9 flow cell, which had consistently lower false positive rates among all assays, a larger spread between FP and TP alignment counts, and detected the target organisms in all 36 test samples. This combination also returned an actionable detection result within 10 min of the library being placed on a flow cell for sequencing for 9 out of 12 unique organisms tested. This contrasted with the LIGTN method (the next best method on R9) which returned only 4 organisms within 15 min. Even given the significantly higher cost of the R9 flow cell compared to FL, the recommendation for a fast, consistent, and high confidence detection method out of the methods considered in this study is still the NATBC method on the R9 flow cell.

Future studies will focus on optimizing this down selected library preparation method in operationally relevant matrices to assess the fidelity of the method for threat organism identification. Overall, the complexity as well as preparation time of the Native Barcoding method fell between the Ligation and 4-primer protocols, but yielded faster and cleaner results from the sequencing data produced. However, the findings described in this study need further validation in operational contexts for decision makers to have greater confidence in the reported calls, and to minimize or eliminate false positives that may have been due to cross contamination from processing multiple samples for the same experiment.

## Figures and Tables

**Figure 1 genes-13-01785-f001:**
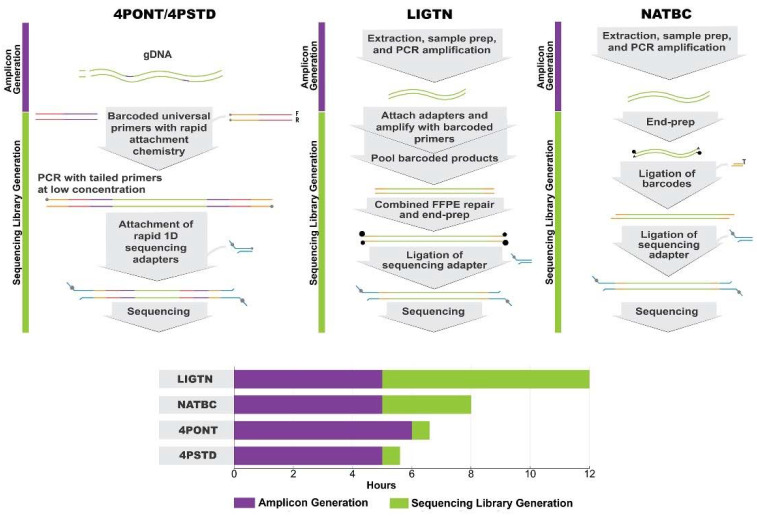
ONT Library Preparation Workflows. Generalized workflow for the four Oxford Nanopore Technology library preparation methods tested. Note that the 4-Primer method covers both the 4PSTD and 4PONT methods, the difference being the PCR cycling parameters. The bar chart displays total time from extracted gDNA to loading onto an ONT flow cell and broken down by the two primary processes: “Amplicon Generation” and “Sequencing Library Generation”: Amplicon Generation includes all steps from gDNA to PCR product, and Sequencing Library Generation includes all steps from PCR product to loading an ONT flow cell.

**Figure 2 genes-13-01785-f002:**
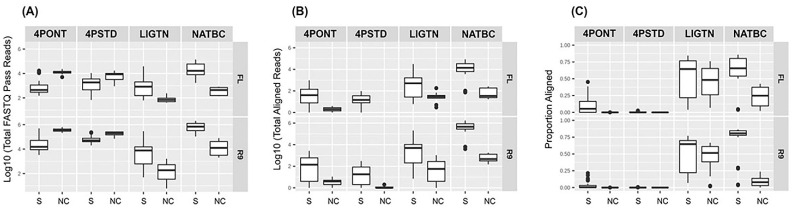
Output read and alignment counts. Read counts from 24 h of sequencing from R9/FL flow cells and the four library prep methods, split into samples spiked with target organism gDNA (0) and NTC (1) groups. (**A**) Total “fastq_pass” read counts (reads with ≥7 average PHRED quality score). (**B**) Total BLASTN aligned “fastq_pass” reads to the amplicon reference database post-filtering (≥90% identity and alignment length). (**C**) The same as (**B**) but displayed as a proportion of (**A**) per method and flow cell type.

**Figure 3 genes-13-01785-f003:**
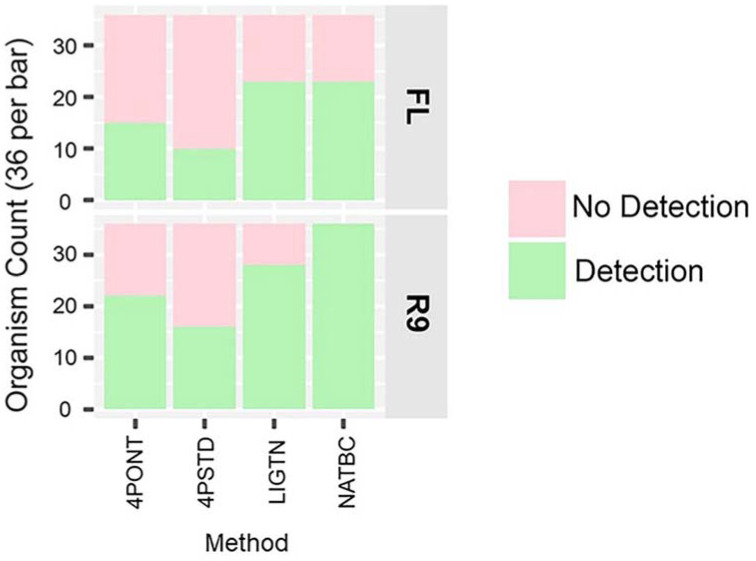
Detected organism counts. Counts for each method out of a total of 36, per method per flow cell type (Green is detected, Pink is not detected). Detection in this analysis means all expected amplicons had a filtered aligned read count ≥1. NTCs are not included.

**Figure 4 genes-13-01785-f004:**
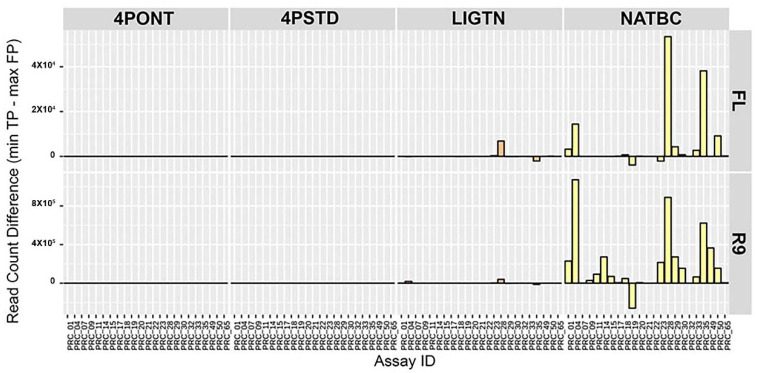
Spread between TP and FP aligned read count per amplicon. Difference between the minimum TP and maximum FP aligned filtered read count per amplicon reference. A larger positive number indicates a greater spread between these two values, and an overall lower FP rate for the assay. NTCs are not included.

**Figure 5 genes-13-01785-f005:**
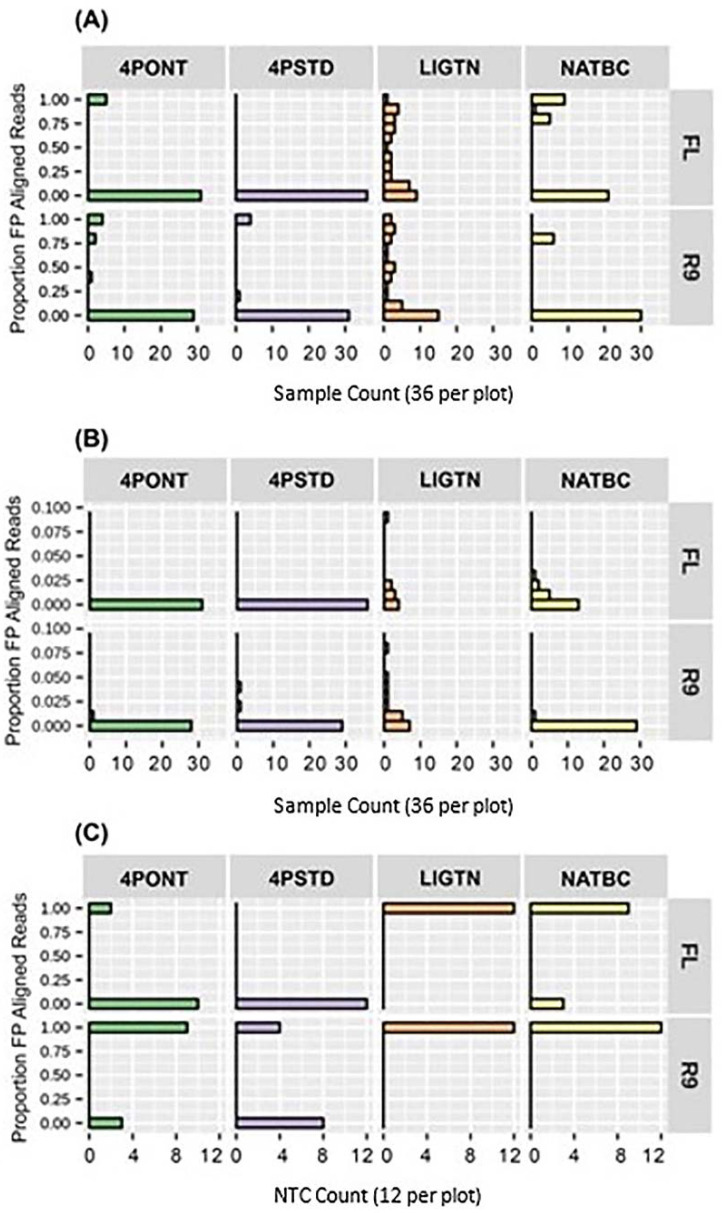
Total FP aligned read counts. Proportion of aligned reads identified as FP after 24 h of sequencing. (**A**) Contains 36 samples per method-flow cell combination, from 0.00 to 1.00 proportion FP. (**B**) A zoom-in of the y-axis of panel (**A**) from 0.00 to 0.10 proportion FP. (**C**) Contains 12 NTCs per method-flow cell combination. Note that panel (**C**) proportions are binary as any aligned read is considered a FP for NTCs.

**Figure 6 genes-13-01785-f006:**
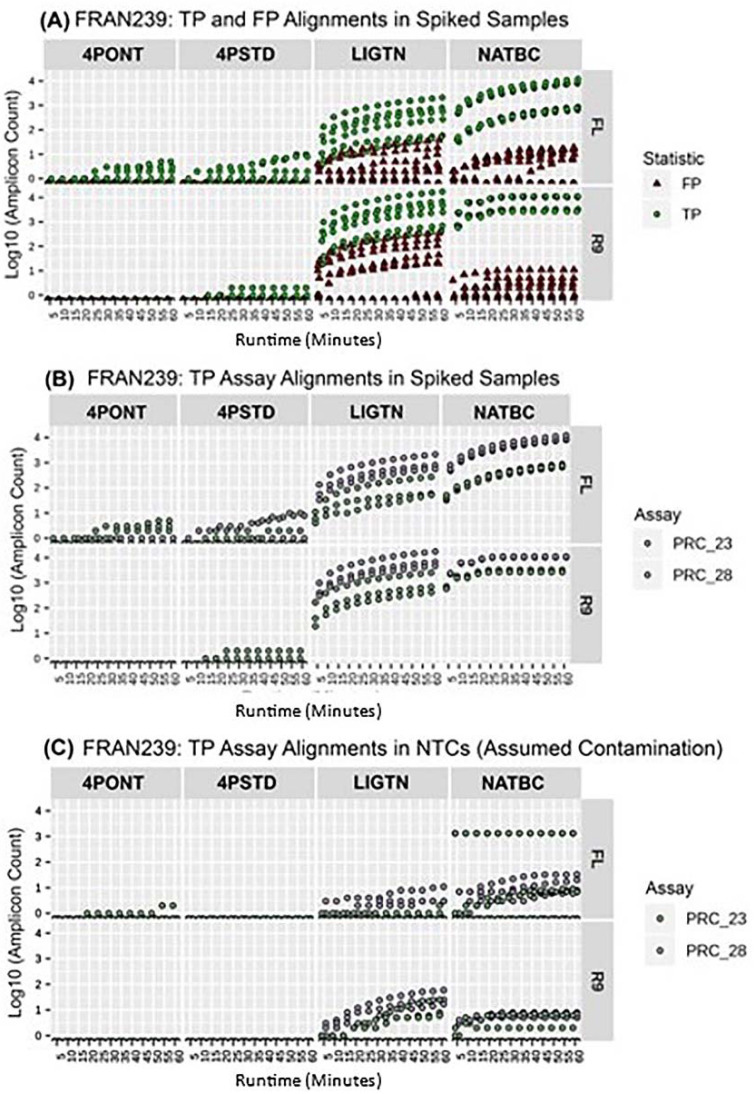
Temporal increases in read counts. The first hour of sequencing for each sample with spiked target-organism gDNA. The amplicon count (y-axis log10) corresponds to the read count of amplicon reference-aligned and filtered reads (≥90% identity and alignment length). The runtime of one hour is split at discrete 5 min intervals, with amplicon counts being a cumulative value from 5 to 60 min. Each method-flow cell type plot represents a sample in biological triplicate. Each replicate then may have amplicon counts for multiple amplicon assays. (**A**) TP and FP assay alignments among the triplicate samples are colored. Does not include NTCs. (**B**) Only TP assay alignments among the triplicate samples are shown and colored for discrimination. Does not include NTCs. (**C**) TP assay (specific to the spiked organism) alignments among all NTC samples.

**Table 1 genes-13-01785-t001:** Overview of ONT library preparation protocols. Four ONT library preparation protocols are shown with major PCR and sequencing library prep steps indicated. An ‘x’ indicates that the corresponding protocol step is required for the method. The various steps for “Sample preparation” and “gDNA extraction” rows under Time are not broken down in this table but would include activities such as sample collection, elution, cell lysis, and DNA extraction.

Method Name		4-Primer ONT	4-Primer Standard	Ligation	Native Barcoding
Method ID		4PONT	4PSTD	LIGTN	NATBC
PCR Step	Standard Targeted PCR			x	x
	4P PCR (ONT cycling parameters)	x			
	4P PCR (custom cycling parameters)		x		
Library Prep Step	End prep and cleanup			x	x
	Ligate barcode adapter and cleanup			x	
	Attach barcode with PCR			x	
	NB ligation and cleanup				x
	Pool barcoded libraries			x	x
	End prep and ligate seq adapter	x	x	x	x
	Load pooled library on to flow cell	x	x	x	x
Time (hours)	Sample preparation	2	2	2	2
	gDNA extraction	2	2	2	2
	PCR amplification	0	0	1	1
	Library prep	2	1	7	3
	Total time until sequencing	6	5	12	8

**Table 2 genes-13-01785-t002:** Organism ID and strain information for PCR assay targets. Organism ID and strain information, along with which PCR assays are expected to amplify are shown, totaling 20 unique assays (PCR primer pairs). There are 3 assays specific for Variola virus included in the 23-plex but not shown in the table because these were not expected to amplify any of the test organisms.

No.	Organism ID	Organism	Strain	Expected PRC Assays
1	BANT708	*Bacillus anthracis*	Sterne BAP708	01, 04, 07
2	BCER248	*Bacillus cereus*	NRS 248	07
3	BRUC105	*Brucella abortus*	RB51	32, 33, 35
4	BRUC106	*Brucella abortus*	Strain 19	32, 33, 35
5	BURK164	*Burkholderia humptydooensis*	MSMB121	49
6	BURK197	*Burkholderia pseudomallei*	JW270	50, 65
7	FRAN239	*Francisella tularensis*	NIH B-38	23, 28
8	FRAN240	*Francisella tularensis*	LVS	23, 29
9	FRAN241	*Francisella tularensis*	Novidica U112	23, 30
10	VACCIN	*Vaccinia*		17, 18, 20
11	YERS113	*Yersinia pestis*	CO92 Lcr (-)	09, 11, 15
12	YERS114	*Yersinia pestis*	CO92 pgm (-)	09, 14, 15

**Table 3 genes-13-01785-t003:** Minimum sequencing time required for TP calls. Minimum sequencing time required for TP calls for (**A**) FL type flow cell and (**B**) R9 type flow cell. TP was defined as >9 reads aligning to the associated amplicon reference in all three replicates per sample.

		Minutes until All TP Amplicons > 9 Reads							
	Device	Flongle Flow Cell				R9 Flow Cell			
	Method	4PONT	4PSTD	LIGTN	NATBC	4PONT	4PSTD	LIGTN	NATBC
Organism ID	BANT708	>60	>60	>60	10	>60	>60	15	5
	BCER248	>60	>60	>60	>60	>60	>60	>60	>60
	BRUC105	>60	>60	>60	>60	>60	>60	>60	>60
	BRUC106	>60	>60	>60	>60	>60	>60	>60	>60
	BURK164	>60	>60	>60	>60	>60	>60	>60	5
	BURK197	>60	>60	>60	45	>60	>60	>60	10
	FRAN239	>60	>60	25	5	>60	>60	5	5
	FRAN240	>60	>60	35	5	>60	>60	10	5
	FRAN241	>60	>60	50	5	>60	>60	10	5
	VACCIN	>60	>60	>60	>60	>60	>60	>60	5
	YERS113	>60	>60	>60	>60	>60	>60	>60	5
	YERS114	>60	>60	>60	>60	>60	>60	>60	10
Min		>60	>60	25	10	>60	>60	5	5
Max		>60	>60	>60	>60	>60	>60	>60	>60

**Table 4 genes-13-01785-t004:** FP aligned read statistics. Mean, median, and standard deviation for total (percent) FP aligned reads among samples [group 0] and total (percent) aligned reads among NTC triplicates [group 1] per method and flow cell type.

Method	4PONT		4PSTD		LIGTN		NATBC		
NTC/Sample	NTC	Sample	NTC	Sample	NTC	Sample	NTC	Sample	
Mean	0 (14)	0 (0)	0 (0)	0 (0)	204 (38)	41 (44)	2547 (39)	70 (22)	FL
Median	0 (0)	0 (0)	0 (0)	0 (0)	26 (25)	30 (48)	247 (2)	24 (25)	
Standard Deviation	1 (35)	1 (0)	0 (0)	0 (0)	598 (35)	47 (24)	4162 (46)	96 (16)	
Mean	1 (17)	3 (0)	0 (12)	0 (0)	1266 (30)	202 (44)	19447 (13)	753 (9)	R9
Median	0 (0)	3 (0)	0 (0)	0 (0)	94 (11)	84 (52)	350 (0)	464 (8)	
Standard Deviation	4 (36)	4 (0)	1 (32)	1 (0)	3343 (36)	309 (24)	63551 (29)	622 (8)	

## Data Availability

The ONT-DART platform is available on APL’s Biological Sciences public Github repository (https://github.com/jhuapl-bio/ONT-DART, accessed on 1 September 2021). Custom script for parsing ONT reads by time period is available on GitHub (https://github.com/raplayer/nanotimeparse, accessed on 1 September 2021).

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
