# Peer review of "Optimization of Oxford Nanopore Technology Sequencing Workflow for Detection of Amplicons in Real Time Using ONT-DART Tool"

_genes, 2022, doi:10.3390/genes13101785_

Round 1
Reviewer 1 Report
If there is a novel contribution here, I am not finding it. I do not see how this paper puts forward a contribution that warrants publication and introduction into the body of scientific literature. In essence, the authors demonstrate that widely adopted genome sequencing technologies behave as they were designed to. In addition, the bioinformatics are rudimentary and it is not clear to me that it is superior to those offered by ONT.
L15 – abstract – is contradictory / wrong.
“Abstract: A targeted genomic sequencing-based high-throughput assay is currently not available 15 for routine biodefense and biosurveillance applications. Earlier, we addressed this gap using a multiplex end-point PCR assay followed by Oxford Nanopore Technology (ONT) based amplicon sequencing and customized data processing.”
First, you state that there is no such high throughput assay, but then say you previously made a high throughput assay for biodefense applications.
L53 – the phrase “metagenome whole genome sequencing (WGS)” is unclear. If one means metagenomics, either include the ‘m’ in the WGS acronym (mWGS), or just call it metagenomics. Whole genome sequencing (WGS) conventionally refers to isolate sequencing.
L58/59 (“however, there have been only a handful of studies investi-58 gating the feasibility of AmpSeq or WGS sequencing approaches in biodefense and clinical 59 applications to date [3-9].”) is misleading and/or incorrect. There is in fact a substantial literature on the usage of nanopore sequencing of both amplicons and metagenomes for these applications, although perhaps least for biodefense applications.
Table 2 has a typo? PCR à PRC
RE the bioinformatics methods: why use blastn? Long read error optimized methods exist for nanopore reads (e.g., minimap2). Further, what was the rationale for the 90% identity cut-off? Can the authors provide guidance on this WRT the intrinsic error rate of nanopore sequencing.
Author Response
Reviewer 1.
If there is a novel contribution here, I am not finding it. I do not see how this paper puts forward a contribution that warrants publication and introduction into the body of scientific literature. In essence, the authors demonstrate that widely adopted genome sequencing technologies behave as they were designed to. In addition, the bioinformatics are rudimentary and it is not clear to me that it is superior to those offered by ONT.
>>We appreciate the candid evaluation of our manuscript and pointed comments by the reviewer. To be fair, we have not claimed novelty in this work and we have clearly laid out the gaps in the scientific literature for this specific area/specific use case, and how we address the gap in this manuscript. In addition, there may have been research studies on AmpSeq in other applications or even in Biodefense but that does not mean that they are available or widely used by the community. To the best of our knowledge, in studies on Amp seq/ Metaseq for biodefense applications, even the basic questions on limit of detection or simple comparison of the real time PCR performance against AmpSeq has not be done. In this study, we are trying to develop an ‘optimized workflow’ and a standardized method for a particular use case. We have tried to articulate that in our previous paper and here as well in the revised version. Again, we are not claiming that we designed these technologies or ‘behave’ as though we are the inventors/ designers. The comment on rudimentary bioinformatics is unfortunate. To give an example, the reviewer probably knows that there are a lot of genome sequence classifiers out there (>165 in one survey) and that they all do the same basic thing; i.e., classify reads, but are unique in some ways. The point is, if one tool is superior and widely adopted then there would not have been a need for all the other tools. Existence of a widely adopted tool, does not preclude others to try different approaches and develop other tools. So, we are not claiming our tool is superior to that offered by ONT. We had started developing ONT-DART way before ONT tools were widely available for this specific use case and aimed to put this on a portable Mk1C-like device. Please keep in mind that we have specific PCR target sequences and libraries to compare to and adjudicate the results- ONT-DART is just one tool for that application as there are other tools.
L15 – abstract – is contradictory / wrong.
>>Thanks for pointing this out. We have changed the current version to more appropriately articulate our goal/objective and the gap.
“
Abstract:
A targeted genomic sequencing-based high-throughput assay is currently not available for routine biodefense and biosurveillance applications. Earlier, we addressed this gap using a multiplex end-point PCR assay followed by Oxford Nanopore Technology(ONT) based amplicon sequencing and customized data processing.”
First, you state that there is no such high throughput assay, but then say you previously made a high throughput assay for biodefense applications.
>>In our previous paper, we established the baseline comparison of real time PCR to amplicon sequencing. In that study, we came to realize the need for optimization of the various steps in the workflow to reduce the time frame to less than 8 hours which is what we have attempted here. We have changed that sentence to be clearer and removed contradictions.
“An optimized, well-tested and validated targeted genomic sequencing-based high-throughput assay is currently not available ready for routine biodefense and biosurveillance applications. Earlier, we addressed this gap by developing and establishing baseline comparisons of a multiplex end-point PCR assay followed by Oxford Nanopore Technology(ONT) based amplicon sequencing to real time PCR and customized data processing”. (Lines 15-18).
L53 – the phrase “metagenome whole genome sequencing (WGS)” is unclear. If one means metagenomics, either include the ‘m’ in the WGS acronym (mWGS), or just call it metagenomics. Whole genome sequencing (WGS) conventionally refers to isolate sequencing.
>>Agreed. We changed the word to MetaSeq to refer to metagenome sequencing and removed any reference to whole genome sequencing accordingly. (Lines 51, 52 and 58).
L58/59 (“however, there have been only a handful of studies investigating the feasibility of AmpSeq or mWGS sequencing approaches in biodefense and clinical applications to date [3-9].”) is misleading and/or incorrect. There is in fact a substantial literature on the usage of nanopore sequencing of both amplicons and metagenomes for these applications, although perhaps least for biodefense applications.
>>We agree with this remark fully and hence completed removed “and clinical applications” (Line 59).
Table 2 has a typo? PCR à PRC
>>It so happens that our internal PRC designations for assays appear to be typo of PCR. PRC is indeed correct, and we have used it in the previously published article.
RE the bioinformatics methods: why use blastn? Long read error optimized methods exist for nanopore reads (e.g., minimap2). Further, what was the rationale for the 90% identity cut-off?
>> The use of blastn relates to history in our lab and development of other tools prior to availability of minimap2. Again, we did not see significant downside to using blatsn as opposed to minimap2 in performance because the database index is exceedingly small. BLASTN also produces more robust statistics regarding each sequence alignment that are leveraged to sort and filter the results.
The rationale for using 90% identity cutoff is based on wet lab testing of real time PCR assays. We have found that templates with 90% match to the primer/probe/amplicon sequences can still produce true positive results and also based on the arbitrary cutoff of identities of target organisms/amplicons to near neighbors.
Can the authors provide guidance on this WRT the intrinsic error rate of nanopore sequencing?
We have used the ONT default PHRED quality score of >7 as the threshold for binning reads into "fastq_pass" and "fastq_fail" bins (note that a conservative average quality estimate for an R9 and FLG flowcell is 12 and 10, respectively). We only use the "fastq_pass" reads in our analysis, and the subsequent identity and alignment length thresholds are then applied to the alignment output from these reads. There are plenty of reads from any given ONT run that will not average a PHRED score of 7 over the length of a read, and there are many failed reads in our runs, especially in the 4-primer methods (that will not meet that threshold). We did check to see how these "fastq_fail" binned reads would map using the same blastn amplicon database and in most cases, no alignments could be made using such low quality reads (note that that blast uses the fasta format as the input anyway, and that if any of the "fastq_fail" reads are in fact derived from amplicons they contained so many positional base call errors that an alignment was not produced by the underlying blast algorithm).

Reviewer 2 Report
GENERAL COMMENTS.
The study aims to benchmark four different Oxford Nanopore library preparation methods using the novel software platform ONT-DART. The sample material consisted of gDNA from known organisms amplified using multiplex primer pools to detect organisms of interest. Benchmarking metrics include the time the four protocols uses and number of TP and FP reads, as well as ability to detect target organisms. The study has two stories, one regarding the benchmarking metrics of the library preparation protocols and another regarding the novel ONT-DART software. Overall, these two stories are not properly intertwined in the article and it seems like it either should be two articles by themselves or in what way ONT-DART improves on similar software that already exists and corrects for FP/FN should be presented earlier in the article. The language of the article leaves a bit to be desired, the text is tedious to read and not structured in a way that seems the most logical. Several terms are also used interchangeably (amplicons, assays, PCR pairs) where in many cases it would be easier to read if the authors stuck to one term. Most of the plots need to be finished (rotate x axis labels, Instead of 0 and 1, write sample/NTC/detected/not detected etc., capital letters for labels on the x and y axis and legends). This criticism extends to tables in the article as well.
Major comments:
- Second paragraph in the introduction (starting line 47) should be rewritten. It is unclear what the authors mean with this paragraph. Amplicon-based sequencing strategies seems to be introduced as “AmpSeq”, but don’t use the term beyond this paragraph. If “traditional PCR” in this case refers to PCR assays and their limitations, it should be changed accordingly. Overall, the paragraph gives the impression of trying to explain the benefits of amplicon-based sequencing assays but fails in doing so. Also, the statement regarding the handful of studies regarding the feasibility of amplicon-based sequencing or WGS in biodefence and clinical applications – this is just not true for the clinical statements, there is a throve of studies investigating different library preparations and sequencing approaches in clinical settings (Typing in "Nanopore Clinical" in google scholar retrieves 13,2000 entries).
- Regarding the description of ONT-DART pipeline – consider a flow chart describing the steps to help the reader. Important for the whole study.
- 3.4: Most of this section describes the ONT-DART pipeline and how it operates. It should be presented in the Methods part and merged with 2.7.
- “Assays” as “PCR pairs / amplicons” are first introduced in table 2 – the terms amplicons, assays, PCR assays are all used to describe amplicons (generated from a pair of PCR primers), this makes the article tedious to read.
- All plots: Use capital letters for x- and y-axis labels and legend. Rotate x-axis tick labels. Don’t use 0 and 1 for the group names, write “sample” and “NTC”.
Minor comments:
-
- Table 1: “PCR with universal adapter tails” does not have any X in the corresponding row. Also “ligate seq adapter”.
- 2.2: It is not entirely clear if the template for the initial amplification for the 4-primer method is using the 23-plex primer pool or not. It is more clearly stated in lines 204-206, but should also be stated in this section.
- 2.3: It would be nice with a sentence for the reasoning behind the changes to the cycling parameters.
- Figure 1 has low resolution
- Table 1: ligate seq adapter.
- Table 2: Expected PRC assays
- Figure 2: Use black text for the plot facet labels.
The list of minor comments is not exhaustive
Author Response
The study aims to benchmark four different Oxford Nanopore library preparation methods using the novel software platform ONT-DART. The sample material consisted of gDNA
from known organisms amplified using multiplex primer pools to detect organisms of
interest. Benchmarking metrics include the time the four protocols uses and number of TP
and FP reads, as well as ability to detect target organisms. The study has two stories, one
regarding the benchmarking metrics of the library preparation protocols and another
regarding the novel ONT-DART software. Overall, these two stories are not properly
intertwined in the article and it seems like it either should be two articles by themselves or
in what way ONT-DART improves on similar software that already exists and corrects for
FP/FN should be presented earlier in the article. The language of the article leaves a bit to
be desired, the text is tedious to read and not structured in a way that seems the most
logical. Several terms are also used interchangeably (amplicons, assays, PCR pairs)
where in many cases it would be easier to read if the authors stuck to one term. Most of
the plots need to be finished (rotate x axis labels, Instead of 0 and 1, write
sample/NTC/detected/not detected etc., capital letters for labels on the x and y axis and
legends). This criticism extends to tables in the article as well.
>> We respectfully disagree with the reviewer’s opinion on how the two stories in our manuscript are intertwined and yet does not flow from one to the other and the suggestion to merge sections 2.7 and 3.4. We have restrained from moving section 3.4 to 2.7 because, even though there is a description of how reads are processed and filtered in the pipeline in 3.4, the description of the pipeline naturally follows with the analysis of the read data in Table 4 and prior sections. We have added bridge sentences (lines 328-330) to clarify this point.
>>We have also added a brief narrative on the unique distinguishing features of ONT-DART in the Introduction section (lines 72-72).
>> We also feel our use of the English language is appropriate and clear enough to fully describe our study in as efficient detail as reasonable possible. The manuscript was read and approved by all authors and others who are native English speakers.
>> We do not use the three terms amplicons, assays, and PCR pairs interchangeably because they are indeed distinct terms: 1) amplicons are the results of a PCR, 2) assays are a general term for the test used to identify the presence/absence of a target region in a DNA sample (as defined by the PCR/primer pair), and PCR pairs are the primer pair sequences used in the PCR that produces a target amplicon.
>> Thanks for pointing out the various issues with figure quality and figure and table labels. We have fixed all those issues.
Major comments:
- Second paragraph in the introduction (starting line 47) should be rewritten. It is unclear what the authors mean with this paragraph. Amplicon-based sequencing strategies seems to be introduced as “AmpSeq”, but don’t use the term beyond this paragraph. If “traditional PCR” in this case refers to PCR assays and their limitations, it should be changed accordingly. Overall, the paragraph gives the impression of trying to explain the benefits of amplicon-based sequencing assays but fails in doing so. Also, the statement regarding the handful of studies regarding the feasibility of amplicon-based sequencing or WGS in biodefence and clinical applications – this is just not true for the clinical statements, there is a throve of studies investigating different library preparations and sequencing approaches in clinical settings (Typing in "Nanopore Clinical" in google scholar retrieves 13,2000 entries).
>> AmpSeq is not used throughout the manuscript because the term “amplicon sequencing” does not appear after the introduction. Generally, we are simply referring to amplicons in the context of sequencing methods and therefore did not have the need to call back to the AmpSeq term in later sections.
>>Yes. Traditional PCR is meant to refer to real time PCR assay and has been chaned accordingly (line 53).
>> We agree with the reviewer’s criticism that there are indeed more than a handful of studies regarding feasibility of AmpSeq for clinical settings and have removed reference to “clinical applications”. We have kept the statement the same otherwise, as this remains true for biodefense applications.
- Regarding the description of ONT-DART pipeline – consider a flow chart describing the steps to help the reader. Important for the whole study.
>> ONT-DART flow chart included as supplemental figure 12.
- 3.4: Most of this section describes the ONT-DART pipeline and how it operates. It should be presented in the Methods part and merged with 2.7.
>> Addressed above.
- “Assays” as “PCR pairs / amplicons” are first introduced in table 2 – the terms amplicons, assays, PCR assays are all used to describe amplicons (generated from a pair of PCR primers), this makes the article tedious to read.
>> As tedious as the terms may be, we feel they are all required to adequately describe the study in enough detail in the context of the sentence to clearly convey the message. All plots: Use capital letters for x- and y-axis labels and legend. Rotate x-axis tick labels.
>> Thanks for this remark and we have addressed this.
- Don’t use 0 and 1 for the group names, write “sample” and “NTC”.
>> Agreed, we have adjusted in all the relevant places in figures and tables.
Minor comments:
- Table 1: “PCR with universal adapter tails” does not have any X in the corresponding row (also “ligate seq adapter”).
>> We have removed the former row entirely from Table 1, but feel it is important to keep the other steps (such as “ligate seq adapter” etc) that occur in all methods.
- 2.2: It is not entirely clear if the template for the initial amplification for the 4-primer method is using the 23-plex primer pool or not. It is more clearly stated in lines 204-206, but should also be stated in this section.
>> We have addressed this by adding the parentheses to section 2.2: “Briefly, an up-front amplification (using a 23-plex primer cocktail) and barcoding step was performed [...]”(Line 97)
- 2.3: It would be nice with a sentence for the reasoning behind the changes to the cycling parameters.
>> We have added “ that are used in current PCR field applications” to the end of the first sentence in section 2.3. (Line 109)
- Figure 1 has low resolution – fixed with a high-resolution image.
- Table 1: ligate seq adapter. - fixed
- Table 2: Expected PRC assays- PRC is not a typo and indeed correct and represents an in internal designation for all PCR assays.
- Figure 2: Use black text for the plot facet labels.- fixed.
- The list of minor comments is not exhaustive
>> We thank the reviewer for the detailed feedback and have addressed these types of minor issues in the revised version.

Round 2
Reviewer 1 Report
Thanks to the authors for a thorough rebut-- my apologies for poor choice of words when writing my critique.
Reviewer 2 Report
I would like to thank the authors for implementing the suggestions.